# Effect of Pulse Repetition Rate on Ultrafast Laser-Induced Modification of Sodium Germanate Glass

**DOI:** 10.3390/nano13071208

**Published:** 2023-03-29

**Authors:** Sergey V. Lotarev, Sergey S. Fedotov, Alyona I. Pomigueva, Alexey S. Lipatiev, Vladimir N. Sigaev

**Affiliations:** Department of Glass and Glass-Ceramics, Mendeleev University of Chemical Technology, Miusskaya pl. 9, 125047 Moscow, Russia; fedotov.s.s@muctr.ru (S.S.F.); pomigyeva_a_i@muctr.ru (A.I.P.); lipatievas@muctr.ru (A.S.L.); vlad.sigaev@gmail.com (V.N.S.)

**Keywords:** sodium germanate glass, nanograting, form birefringence, laser-induced crystallization, ultrashort laser pulses

## Abstract

We report an unexpected pulse repetition rate effect on ultrafast-laser modification of sodium germanate glass with the composition 22Na_2_O 78GeO_2_. While at a lower pulse repetition rate (~≤250 kHz), the inscription of nanogratings possessing form birefringence is observed under series of 10^5^–10^6^ pulses, a higher pulse repetition rate launches peripheral microcrystallization with precipitation of the Na_2_Ge_4_O_9_ phase around the laser-exposed area due to the thermal effect of femtosecond pulses via cumulative heating. Depending on the pulse energy, the repetition rate ranges corresponding to nanograting formation and microcrystallization can overlap or be separated from each other. Regardless of crystallization, the unusual growth of optical retardance in the nanogratings with the pulse repetition rate starting from a certain threshold has been revealed instead of a gradual decrease in retardance with the pulse repetition rate earlier reported for some other glasses. The repetition rate threshold of the retardance growth is shown to be inversely related to the pulse energy and to vary from ~70 to 200 kHz in the studied energy range. This effect can be presumably assigned to the chemical composition shift due to the thermal diffusion of sodium cations occurring at higher pulse repetition rates when the thermal effect of the ultrashort laser pulses becomes noticeable.

## 1. Introduction

Nanogratings, often referred to as Type-II-fs modifications, are self-organized bulk nanoperiodical structures often regarded as the most intriguing ultrafast laser-induced phenomena in dielectric materials. This type of micromodification has become the first example of a regular sub-wavelength structure inscribed by optical methods inside homogeneous dielectrics [1]. A detailed investigation of the structure of nanogratings in fused silica showed that they consisted of parallel nanolayers (“nanoplanes”) possessing a reduced density and refractive index because of the presence of oxygen-filled nanopores separated with thicker layers of solid glass [2]. Due to nanoperiodic refractive index variation, they exhibit form birefringence similar to that of a uniaxial negative crystal. Its optical axis orientation and induced retardance can be controlled by the determination of laser writing parameters. Controllable birefringence with extraordinary thermal stability [3] and reduced chemical durability [4] of nanogratings enabled their application for ultrastable optical data storage, light polarization converters with patterned birefringence and microfluidic circuits [5]. Numerical simulation showed a crucial role of laser-induced defects in launching the self-assembly process [6,7].

Unlike the laser-induced periodic surface structures often referred to as ripples shown to be a universal phenomenon a long time ago [8,9], nanogratings as a bulk modification emerging in the inside of materials were discovered and widely investigated over a decade only in pure and doped fused silica [1,2,5,10]. Later, nanogratings were obtained in vitreous GeO_2_ [11,12] and in a number of binary and multicomponent glasses [13,14,15,16,17,18,19,20,21,22,23,24,25,26,27,28], including multicomponent borosilicate Borofloat 33 and BK7 (Corning) [13,14,15], alkali-free/depleted aluminoborosilicate AF32 (Schott) [14], 7059, Eagle XG (Corning), and soda lime [15] glasses, binary titanium silicate ULE (Corning) [13], Na_2_O-SiO_2_ [16,17], GeO_2_-SiO_2_ [18,19], Al_2_O_3_-SiO_2_ [20], Na_2_O-GeO_2_ [21,22] and ternary Na_2_O_3_-B_2_O_3_-SiO_2_ [23] glasses. Nanogratings having an atypically large period were recently obtained in borate glasses, namely in sodium borate and sodium aluminoborate systems [24,25]. A specific type of nanograting is based on the laser-induced phase separation process resulting in the precipitation of nanoperiodical crystalline layers. It was obtained in glasses free of conventional glass-forming oxides: Dy_2_O_3_-Al_2_O_3_ [26,27] and La_2_O_3_–Ta_2_O_5_–Nb_2_O_5_ [28] glasses. Similar nanogratings formed by layers of oriented LiNbO_3_ nanocrystals were shown in lithium niobium silicate [29,30] and lithium niobium borosilicate [31] glasses. Appearing to be a common phenomenon for oxide dielectrics, nanogratings were also inscribed in single crystals such as quartz [32], sapphire [33] and Nd:YAG [34]. In the case of YAG (Y_3_Al_5_O_12_) crystal, their formation was shown to originate from the phase transformation rather than nanopore emergence, and the nanoplanes were formed via Y_3_Al_5_O_12_ amorphization or recrystallization into a perovskite-like phase depending on laser writing conditions [34].

In contrast to crystals, glass composition can be varied gradually, which allows for the gradual variation of its properties. Investigation of the effect of glass composition on the formation and properties of nanogratings is now in progress. Some specific compositional and laser exposure parameter effects on nanograting formation in multicomponent glasses [13,15,16,18,19,20,21,22] distinguishing them from fused silica have been revealed but not yet fully explained. Recently, we have shown the formation of nanogratings in a series of binary sodium germanate (SG) glasses containing from 3 to 22 mol.% Na_2_O [22]. Interestingly, the formation of nanogratings in 22Na_2_O·78GeO_2_ glass at the applied laser exposure conditions was accompanied by the precipitation of Na_2_Ge_4_O_9_ microcrystals around them. In the present study, we examined the effect of the laser pulse repetition rate (PRR) on the ultrafast laser-induced modification of 22Na_2_O·78GeO_2_ glass (hereafter referred to as SG22) in the 50 kHz–1 MHz range.

## 2. Materials and Methods

Glass under study was fabricated from chemically pure Na_2_CO_3_ and GeO_2_. The batch was molten in a platinum crucible in the electric furnace for 1.5 h at 1100 °C and the melt was quenched between two steel plates. The obtained glass sample had a density equal to 3.94 ± 0.01 g/cm^3^ and glass transition temperature T_g_ = 516 °C. More details on its synthesis and characterization were reported earlier [22]. The as-quenched glass was annealed for 8 h at T_g_-30 °C and then cut into ~1 mm thick rectangular plane parallel plates, four sides of which were polished to a mirror grade to provide the possibility of top and side view observation.

Pharos SP Yb:KGW-based regeneratively amplified femtosecond laser (Light Conversion Ltd., Vilnius, Lithuania) generating pulses with a variable duration from 180 fs to 10 ps at a tunable PRR up to 1 MHz at 1030 ± 2 nm wavelength was applied for modification of the fabricated glass samples. The sample was positioned using an Aerotech ABL1000 XYZ-motorized air-bearing stage (Aerotech Inc., Pittsburgh, PA, USA ). A layout of the setup can be found elsewhere [22]. Sets of dots were written in the glass samples by the stationary laser beam focused into the sample using an Olympus LCPLN50XIR microscope objective (Olympus, Tokyo, Japan) (N.A. = 0.65) at a depth of 50 μm under the glass surface. In the first set of the performed experiments, two pulse energy values within the range based on the previous experiments on SG22 glass [22] were applied: 120 nJ and 210 nJ (if measured at the sample surface), the pulse duration was set to 1.2 ps. The PRR was varied from 67 kHz to 1 MHz. Two mutually orthogonal orientations of the polarization plane of the incident laser beam were used to check the polarization sensitivity of the laser-induced birefringence. In further experiments, we varied the pulse energy from 50 to 250 nJ and the pulse duration from 200 fs to 2 ps. An Olympus BX61 optical microscope (Olympus, Tokyo, Japan) equipped with an Abrio Microbirefringence system (CRi Inc., Woburn, MA, USA) was used in transmission mode for the observation of the laser-written dots and quantitative birefringence microanalysis in top and side view configurations. Numerical mapping of the orientation of the slow axis of birefringence and the optical phase retardance was performed within the observed area. Confocal Raman spectroscopy of some laser-written domains was performed using a Nanofinder confocal Raman spectrometer included in the NTEGRA Spectra facility (NT-MDT Ltd., Moscow, Russia) using a blue emission line (488 nm wavelength) of an Ar ion laser as an excitation source. Atomic force microscopy (AFM) by means of the NTEGRA Spectra facility was used in the contact mode for visualization of the nanoperiodical structure of the laser-written nanogratings. To expose cross-sections of the laser-written dots at different depths, a series of birefringent dots was written in the same laser exposure mode with a slight shift of the focusing depth from dot to dot. Then the glass sample was polished to the level of their inscription using an aqueous suspension of CeO_2_ powder and then the exposed surface was rinsed with 96 vol.% water solution of ethanol for 1 min according to the sample preparation technique suggested by Wang et al. [21] and used in our previous study [22].

## 3. Results

The slow axis of birefringence induced in the laser-written dots was perpendicular to the polarization plane of the writing laser beam up to 250 kHz PRR for both pulse energy values applied (Figure 1, top view), which indicated the formation of nanogratings at these conditions. The dots inscribed at a low PRR are not shown in Figure 1 as they look identical to the dots inscribed at 111 kHz in the case of 210 nJ pulses and at 167 kHz in the case of 120 nJ pulses. A side view pseudocolor map of the slow axis of birefringence in Figure 1 highlights only one orientation because in the other case, the optical axis of uniaxial birefringence (fast axis, in this case) is perpendicular to the image plane; thus, it induces no retardance and cannot be detected in this configuration. At higher pulse energy (210 nJ), 250 kHz PRR appears to be very close to the upper limit value of nanograting formation, as only one of two orthogonal orientations of the writing laser beam induced form birefringence.

At higher PRR, the increasing thermal effect of the femtosecond pulses and growing average temperature of the laser-exposed area during the exposure process must have caused glass melting, which prevented the formation of nanogratings. At the same time, ring-shaped birefringent microdomains whose orientation did not correlate with the laser beam polarization plane can be seen in the laser-written dots in the top view configuration starting from 167 kHz at 210 nJ pulse energy (Figure 1a) and from 333 kHz at 120 nJ pulse energy (Figure 1b). These microdomains were found to be Na_2_Ge_4_O_9_ microcrystals, which was confirmed by confocal Raman spectroscopy (Figure 2). The obtained Raman spectra are very similar to those reported for the laser-induced Na_2_Ge_4_O_9_ crystals in SG22 glass [22] and are omitted in this paper. The laser exposure at 1 MHz PRR produced a number of microcracks together with microcrystallization at both pulse energies.

Mapping of the scattering intensity in the Raman shift range of 510–575 cm^−1^ including the dominant Raman peak of Na_2_Ge_4_O_9_ crystal showed that the crystallized regions had the shape of a tube with a circular cross-section oriented along the beam propagation direction. The peripheral character of crystallization was also confirmed by the AFM images (Figure 3). The ring-shaped bulge at the exposed surface (Figure 3b,c) corresponds to the cross-section of the crystallized region, which appears to be more durable to polishing or to liquid etching than surrounding glass [22]. The width of the crystalline layer varied from 0.4 μm to 0.8 μm for the given laser writing mode according to the AFM images. The lateral precipitation of the crystalline phase relative to the laser beam propagation axis is typical for ultrafast laser-induced crystallization, which is known to start at the boundary of glass and liquid [35]. The PRR range as high as hundreds of kHz is also considered desirable to provide sufficient average temperature for microcrystal growth under the pulses with tens to hundreds of nJ of energy, though at higher pulse energy, crystal growth can be launched even below 10 kHz PRR [36]. It should be noted that at a low repetition rate (Figure 1a, 111–143 kHz), regions with form birefringence induced by 210 nJ pulses are longer, whereas at a higher repetition rate, form birefringence appeared in the upper part of the laser-modified region, farther from the focal point. At the same time, crystallization took place in its lower part, closer to the focal point. A crystallized domain and a nanograting showed only partial overlapping along the beam propagation direction, i.e., along the vertical axis in our case (Figure 3). Evidently, in the latter case, the higher laser beam intensity near the focal point induces glass melting, while at the opposite edge of the modified region, the glass viscosity is still large enough to keep the emerging self-assembled structure of the nanograting.

A detailed quantitative analysis of the laser-induced form birefringence in fused silica, some commercial multicomponent glasses (Schott Borofloat 33, Corning ULE) and vitreous GeO_2_ subjected to the PRR performed by Lancry et al. [37] revealed a monotonous fall of retardance. It gradually accelerated, with PRR starting from 50 to 100 kHz for fused silica and ULE, or from 10 kHz for lower-melting Borofloat 33 and GeO_2_, whereas lower PRR had no effect on the retardance. This was explained by a decrease in the number of pores forming nanoplanes in the nanogratings due to the viscoelastic flow of glass occurring when averaged laser processing temperatures approached the glass softening point. A similar trend was shown by Xie et al. [15] using the example of Borofloat 33 glass in the 10–1000 kHz PRR range and in the full range of pulse energy corresponding to nanograting formation. It was also demonstrated that the lower pulse energy threshold of nanograting formation was independent of the PRR. However, the pulse energy window allowing for nanograting formation gradually decreased with a PRR due to the increase in the thermal effect of laser pulses. The similar pulse energy window behavior was described in detail for nanogratings inscribed in fused silica [38]. In our case, the retardance measured in fused silica remained independent of the PRR up to 500 kHz, likely because of much smaller pulse energy (120 or 210 nJ vs. 1 μJ pulses applied by Lancry et al. [37] at similar focusing conditions) and a less pronounced thermal effect. However, the retardance of the laser-written birefringent domains in SG22 glass, though constant at a low PRR, manifested an unexpected substantial increase with the PRR starting from 110 to 125 kHz (Figure 4), which appears to contradict the abovementioned data reported for various glasses [37]. Data obtained for form birefringence inscribed by 120 nJ and 210 nJ pulses show that higher pulse energy evidently induces an increase in retardance starting from a lower PRR.

To check the effect of the retardance increase in the nanogratings starting from a certain PRR of the beam used for their inscription across a wider pulse energy range, a set of birefringent dots was written by 10^6^ pulses with a 1.2 ps duration and energy varying from 50 to 260 nJ (Figure 5). The accelerating growth of the retardance in the birefringent dots starting from a certain PRR level can be observed at all applied pulse energy values. The corresponding threshold interpulse interval grows monotonously and almost linearly with the pulse energy in the examined range. An instability of retardance at high PRR values corresponding to 2 μs and 3 μs interpulse interval is related to the strong contribution of crystal- and stress-related birefringence together with the degradation of nanogratings. Obviously, the domains written at 500 kHz PRR (2 μs interpulse interval) contain nanogratings only in the case of the relatively low pulse energies of 50 nJ and 80 nJ.

We have also analyzed the dependence of retardance of nanogratings inscribed in SG22 glass on the pulse duration. A 2D pseudocolor plot of retardance in nanogratings written in SG22 glass by and 10^6^ pulses per dot at 210 nJ pulse energy as a function of the pulse duration and the PRR is given in Figure 6. Importantly, an earlier reported [22] sharp pulse duration threshold occurs fully independent of the PRR in the range corresponding to the nanograting formation. The dependence of retardance on the pulse duration, including its fast rise until ~1 ps and further slow fall, is most pronounced for the high retardance obtained at 200 kHz PRR, which also conforms to the previously published data [22]. The slow fall of retardance with the pulse duration for the pulses longer than ~1.2 ps is related to the reduction in the peak intensity of longer laser pulses at the constant pulse energy obviously resulting from a lower nonlinear absorption rate. A similar trend can be observed in the data earlier reported for nanogratings in other glasses, including fused silica, AF32 and Borofloat33 [14].

## 4. Discussion

To explain the unexpected effect of retardance growth with the PRR in the laser-written nanogratings in SG22 glass, earlier reported data on redistribution of alkali cations during the inscription of nanogratings in 15Na_2_O·85SiO_2_ glass are helpful [17]. The formation of those nanogratings using 10^6^–10^7^ pulses with 600 fs duration at 200 kHz PRR was accompanied by the migration of most of Na^+^ cations out of the nanograting driven by the thermal diffusion (also known as Soret effect [39]). The rest of the Na^+^ was concentrated in the nanoplanes, giving rise to the precipitation of Na_2_Si_4_O_9_ nanocrystals. In the germanate glass matrix, the migration of Na^+^ cations out of the focal area of the stationary ultrafast laser beam due to thermal diffusion in the germanate glass matrix was demonstrated for 15Na_2_O·85GeO_2_ glass exposed to 1.4 μJ pulses at 250 kHz PRR [40]. The effect of a nanoperiodical redistribution of remaining sodium into nanoplanes within nanogratings written in SG22 glass can also be expected from the analogy with sodium silicate glass.

Moreover, the higher the content of alkaline oxide in binary silicate [16] or germanate [22] glass, the higher the number of pulses required to induce form birefringence and, in the case of sodium germanate glasses [22], the lower retardance achieved, other conditions being equal. Therefore, sodium depletion inside the nanograting was expected to increase with the PRR due to the intensification of thermal diffusion, when the thermal effect of the laser pulses emerged. A corresponding shift of glass composition in the nanograting to the higher content of GeO_2_ meant the possibility to achieve higher retardance under the same exposure conditions. It should be mentioned that the retardance behavior at various PRR levels reported earlier for various glasses [15,37] was studied in nanogratings written by the moving laser beam at scanning speeds providing 10^3^ pulses/μm overlapping rate. Thus, depending on the pulse energy, the total incident light energy dose applied in those studies occurred to be two or three orders of magnitude lower than in the present experiments in SG22 glass where dots were written by 5·10^5^ or 10^6^ pulses. This can be another reason for such a significant difference in retardance trends, because the introduced energy dose must have been too small for glass composition shift and possible compositional effects on retardance, even in the case of Borofloat 33 glass that contains relatively mobile alkaline ions [37]. In contrast, the total dose of the incident light energy applied for a vivid demonstration of significant sodium migration out of the irradiated area of sodium germanate glass [40] was by two orders of magnitude higher in our case, though less tight focusing partially compensated this difference in terms of the area density of this dose.

Though the role of nanoperiodical alkali cation redistribution inside a nanograting in the rise of form birefringence is not well understood so far, it can also be assumed to give a contribution into the nanoperiodical refractive index contrast providing form birefringence of the nanograting. This cation redistribution process, which can be regarded as a kind of regular phase separation, is also expected to be enhanced by the thermal action of femtosecond pulses, since an increase in the time-averaged temperature facilitates the diffusion of cations. If this contribution into the refractive index contrast is positive in the case of SG22 glass, this can be another reason giving rise to the unusual growth in the laser-induced retardance with the PRR of the writing laser beam.

Significantly, qualitative evaluation of form birefringence in unconventional glasses [27,28] revealed a PRR threshold of about 100 kHz or higher depending on the pulse energy. Below this threshold, crystallized nanogratings cannot be formed due to the necessity of the thermal effect of the ultrafast laser beam for phase separation and nanocrystallization. This principally distinguishes them from nanopore-based, fully amorphous nanogratings in fused silica, ULE glass or Borofloat 33 glass [13,35], which manifest no low PRR threshold for nanograting inscription. Nanogratings formed in SG22 glass could occur in an intermediate case between these two utmost kinds of nanograting realization. The contribution of different mechanisms to a raise of form birefringence in them probably varies depending on the beam parameters. Further detailed analysis of their morphology and elemental distribution is required to confirm this assumption and explain the revealed features.

## 5. Conclusions

In conclusion, ultrafast laser-induced micromodification of 22Na_2_O·78GeO_2_ glass including nanograting formation and microcrystallization is examined and an unexpected PRR effect on ultrafast-laser nanostructuring of 22Na_2_O·78GeO_2_ glass is demonstrated. At a lower PRR (≤250 kHz), the inscription of nanogratings possessing form birefringence is observed under series of 10^5^–10^6^ pulses, whereas a higher PRR launches peripheral microcrystallization with precipitation of Na_2_Ge_4_O_9_ due to the thermal effect of femtosecond pulses. Depending on the laser pulse energy, the PRR ranges corresponding to nanograting formation and crystallization can overlap or be separated from each other. We revealed the unusual growth of optical retardance of the laser-written nanogratings with the PRR, starting from a threshold value with an order of magnitude of ~10^2^ kHz and inversely proportional to the pulse energy, varying from 67 to 200 kHz in the studied range of the pulse energy. This growth gradually accelerates with the pulse repetition rate until the thermal effect of laser pulses is too large for nanograting formation due to glass melting. The optical retardance increase occurs whether crystallization around the nanograting is induced or not and distinguishes the investigated glass from fused silica and some multicomponent glasses studied earlier, which manifest stability or a monotonous decrease in retardance in nanogratings with the PRR. The observed phenomenon is likely related to the thermal diffusion of sodium cations rising with the thermal effect of the laser beam when the thermal effect of the ultrashort laser pulses becomes noticeable.

## Figures and Tables

**Figure 1 nanomaterials-13-01208-f001:**
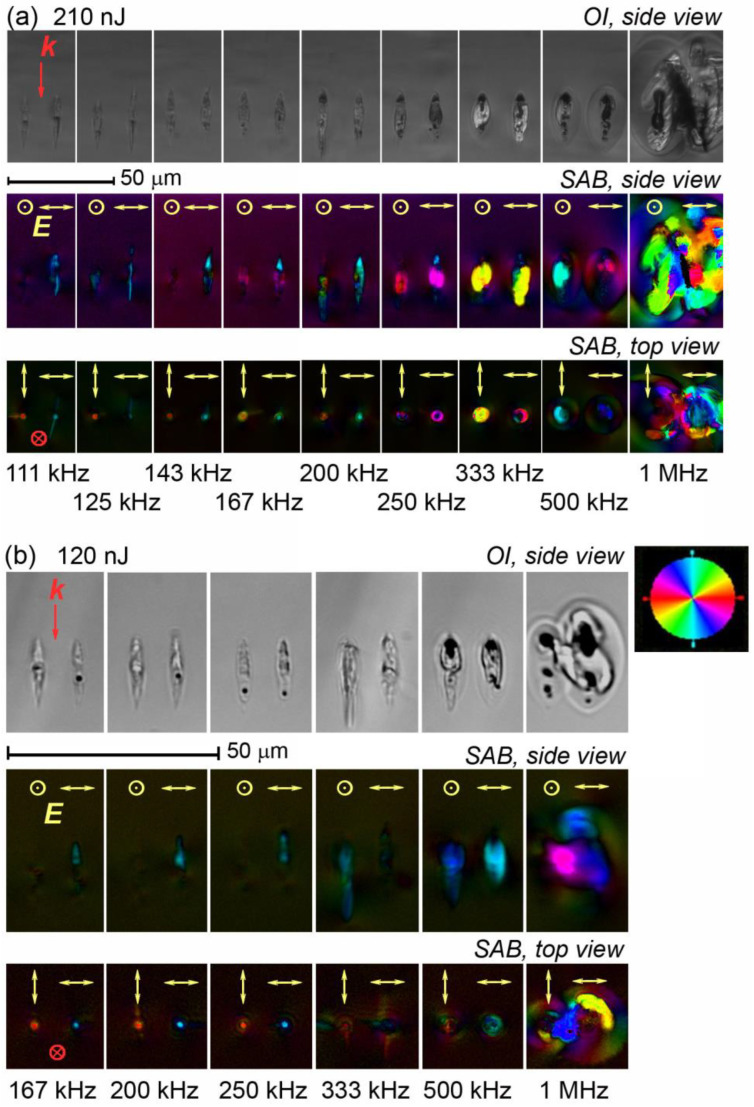
Brightfield optical images (OI) of the side view and pseudocolor maps of the orientation of the slow axis of birefringence (SAB) of the dots written in SG22 glass by 10^6^ laser pulses per dot at the pulse duration of 1.2 ps and the pulse energy of 210 nJ (**a**) and 120 nJ (**b**) and the varied PRR and polarization plane orientation ***E*** (indicated in yellow). The notation ***k*** (indicated in red) is for the wave vector direction.

**Figure 2 nanomaterials-13-01208-f002:**
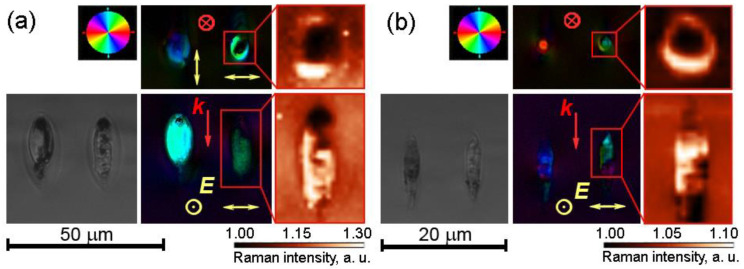
Brightfield OI of side view, pseudocolor maps of the orientation of SAB and Raman maps of nanogratings inscribed in GS22 glass by 5·10^5^ laser pulses with 210 nJ energy at the PRR of 333 kHz (**a**) and 200 kHz (**b**). Top view and side view are shown in the upper and lower rows, respectively. Brightness shows the area under Raman spectra in the 510–575 cm^−1^ range, including the strongest characteristic peak of Na_2_Ge_4_O_9_ crystalline phase. ***k*** and ***E*** indicate writing laser beam propagation direction and polarization plane orientation, respectively.

**Figure 3 nanomaterials-13-01208-f003:**
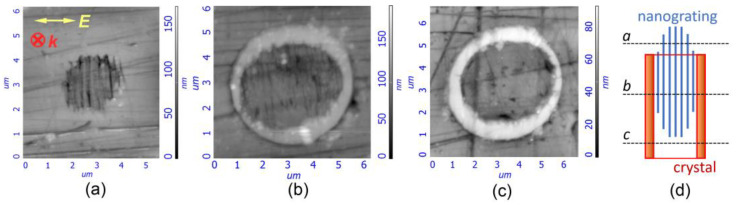
AFM images of the cross-sections of the laser-written birefringent microdomains exposed at a different depth and showing a nanograting (**a**), a nanograting surrounded by crystals (**b**) and a crystalline ring without a nanograting inside it (**c**). A scheme of the cross-section locations relative to the laser-written dot (**d**). The images are acquired from different microdomains written at the same conditions by 10^6^ pulses having 210 nJ energy, 1.2 ps duration and 143 kHz PRR. ***k*** and ***E*** indicate writing laser beam propagation direction and polarization plane orientation, respectively, and are the same for images (**a**–**c**).

**Figure 4 nanomaterials-13-01208-f004:**
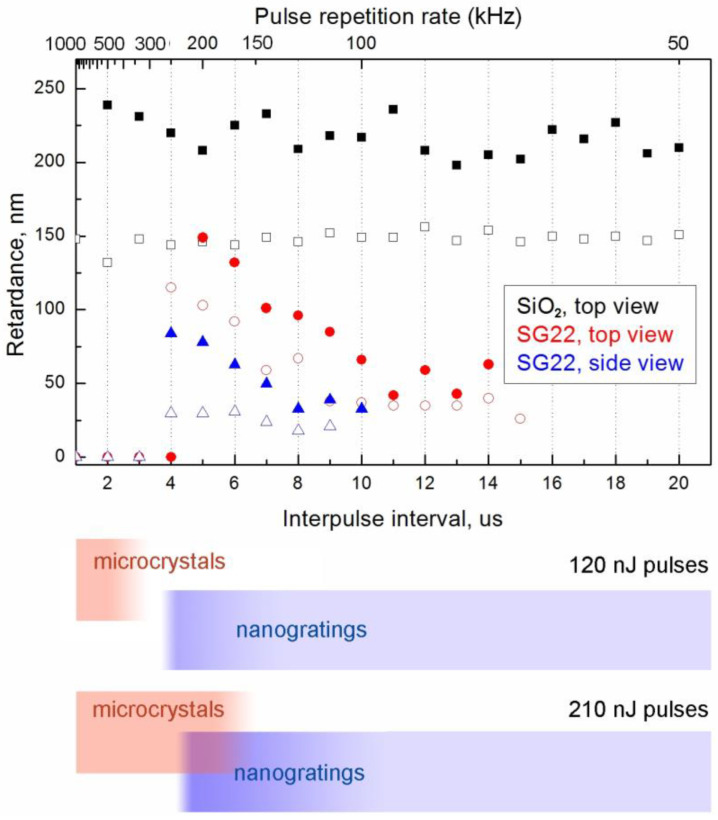
Retardance of nanogratings written by 10^6^ pulses per dot in SG22 glass at 1.2 ps pulse duration, and 120 nJ (○—top view, △—side view) and 210 nJ (●—top view, ▲—side view) pulse energy and in SiO_2_ glass (top view) at 120 nJ (□) and 210 nJ (∎) pulse energy as a function of PRR (**top**). Diagram of the nanograting formation and crystallization at an indicated pulse energy and the same PRR scale (**bottom**).

**Figure 5 nanomaterials-13-01208-f005:**
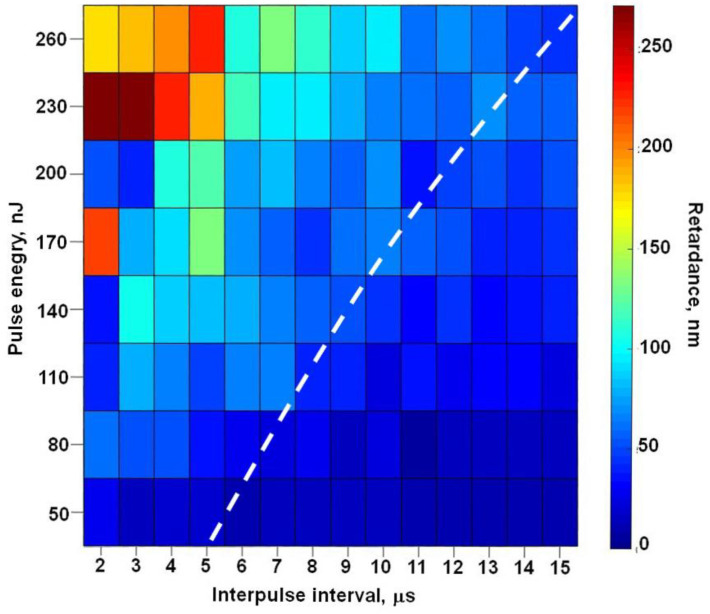
A pseudocolor chart of the retardance in birefringent domains written in SG22 glass by 10^6^ laser pulses of 1.2 ps duration as a function of the pulse energy and the interpulse interval. The white dashed line indicates an approximate boundary at which the measured retardance starts growing over its initial level independent of PRR.

**Figure 6 nanomaterials-13-01208-f006:**
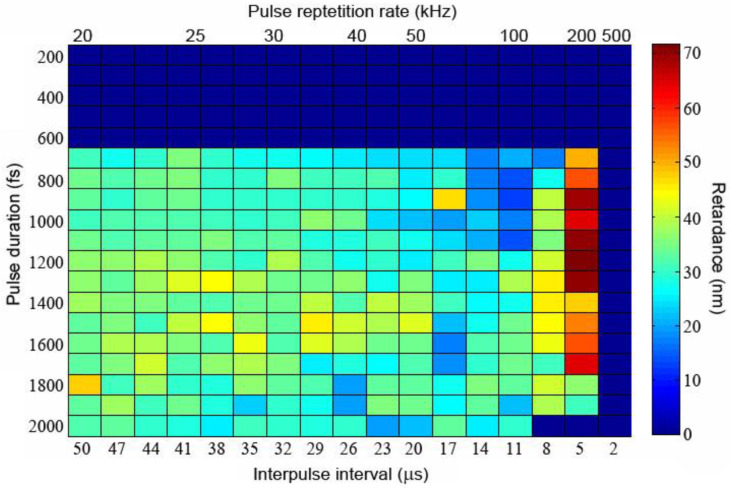
Pseudocolor chart of retardance in nanogratings laser-written in SG22 glass using 10^6^ pulses per dot at 210 nJ pulse energy as a function of pulse duration and PRR.

## Data Availability

Not applicable.

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
