# Peer review of "Effect of Pulse Repetition Rate on Ultrafast Laser-Induced Modification of Sodium Germanate Glass"

_nanomaterials, 2023, doi:10.3390/nano13071208_

Round 1

Reviewer 1 Report

This paper explains about the pulse repetition rate effect on ultrafast-laser modification of glass. The repetition rate is varied between 70 to 200 kHz in the studied energy range. The manuscript is well-written and provides scientific clarity. But I felt the images are not good enough to show the results. Instead of optical microscope images, you could provide some SEM images at least the parts where you see the different effects like birefringence and micro crystallization in the nano gratings. They mentioned the findings are well explained but it will be good if there are clear images to show the observation. 

Here are my comments:

Fig.1 showed that for fixed energy you varied the different PRR, but how much was the pulse duration if it was fixed? If it's varied how do you filter the effect of pulse duration from the observation?

Did you find the ablation threshold for this glass at a different PRR? What is the effect of the repetition rate on the ablation threshold?

It will be good to include some good images of the effects.

I accept the manuscript if you the author could address these minor comments. 

Author Response

Response to Reviewer #1

Reviewer #1: This paper explains about the pulse repetition rate effect on ultrafast-laser modification of glass. The repetition rate is varied between 70 to 200 kHz in the studied energy range. The manuscript is well-written and provides scientific clarity. But I felt the images are not good enough to show the results. Instead of optical microscope images, you could provide some SEM images at least the parts where you see the different effects like birefringence and micro crystallization in the nano gratings. They mentioned the findings are well explained but it will be good if there are clear images to show the observation.

We are grateful to the reviewer for taking the time to consider our manuscript and give helpful recommendations. Revised and newly added fragments of the text are highlighted in “Track Changes” mode.

We agree that SEM images would be very helpful to examine the observed laser-induced effects. However, this sodium germanate glass is a rather difficult object for electron microscopy methods. Due to its high crystallization ability and rather low softening temperature, it easily degrades, changes its structure or crystallizes under the probe electron beam, so it’s not easy to get a reliable relevant picture of the laser-written structure. In particular, this is why we used atomic force microscopy rather than electron microscopy to visualize the submicron structure of the laser-written domains in this and the previous studies. Electron microscopy analysis is planned and now in progress but it may take some noticeable time and efforts to find the proper conditions. We believe it could be a topic of another publication.

On the other hand, the applied combination of methods, including high-resolution polarizing optical microscopy together with confocal Raman spectroscopy, provides an understanding of the main morphological features of the elements of the laser-written domains, while atomic force microscopy gives information of their submicron structure.

Reviewer #1: Here are my comments:

Fig.1 showed that for fixed energy you varied the different PRR, but how much was the pulse duration if it was fixed? If it's varied how do you filter the effect of pulse duration from the observation?

In the experiments illustrated by Fig. 1, the pulse duration was fixed and equal to 1.2 ps, which was mentioned in line 86 of the manuscript: “In the first set of the described experiments, two pulse energy values within the range based on the previous experiments on SG22 glass [19] were applied: 120 nJ and 210 nJ (if measured at the sample surface), the pulse duration being set to 1.2 ps”.

Now we also added it to the caption of Fig. 1 for better clarity.

Reviewer #1: Did you find the ablation threshold for this glass at a different PRR? What is the effect of the repetition rate on the ablation threshold?

It will be good to include some good images of the effects.

This study concerned only volume modification effects, including the formation of nanogratings which are bulk objects and spatially selective bulk microcrystallization. We emphasized it in the reviewed manuscript. These modifications were written by the tightly focused laser beam whose focus was deep enough under the glass surface, so that the surface remained unaffected. Thus we didn’t analyze the surface effects of the laser beam and in particular, the ablation threshold for this glass because it seems to be far out of the scope of the phenomena examined here and requires separate investigation and analysis.

However, if the reviewer and the editor consider its important to be added to to this paper, we can perform this experiment, which would take some additional time.

Reviewer 2 Report

In this paper, the authors investigated the influence of the repetition rate during ultrafast pulsed laser processing of sodium germanate glass (with the composition 22Na2O·78GeO2). Although this technique of nanogratings inscription is well-known in the ultrafast laser processing community, and the paper could be considered a continuation of Ref 19 (Lotarev, S.V.; Fedotov, S.S.; Kurina, A.I.; Lipatiev, A.S.; Sigaev, V.N. Ultrafast laser-induced nanogratings in sodium germanate 341 glasses. Opt. Lett. 2019, 44, 1564–1567), the manuscript presents some new contents and interesting observations. The authors have demonstrated how pulse repetition rate influence the formation of nanogratings and microcrystallization inside glass for two laser energies.

It is reported an unusual growth of optical retardance of the laser-written nanogratings with the PRR which is inversely proportional to the pulse energy. This observation is in opposition with other studies on SiO2 and multicomponent glasses. Why? This effect is explained (hypothetically) as a consequence of the thermal diffusion of sodium cations at high repetition rate of the laser pulses, but not demonstrated. A quantitative analysis of these regions by XPS and/or EDXS mapping is needed to clarify this supposition.

The paper is clear and well-written, however, I have some minor observations to be addressed:

- Most of the experiments (except those devoted to the influence of the pulse duration), were performed with laser pulses of 1.2 ps, but described as femtosecond laser pulses. Please clarify this point.

-  In the caption of Figure 4, „and 210 nJ” should be erased.

- There are some small English mistakes in the manuscript. Please check and correct them.

Author Response

Response to Reviewers

Reviewer #2: It is reported an unusual growth of optical retardance of the laser-written nanogratings with the PRR which is inversely proportional to the pulse energy. This observation is in opposition with other studies on SiO2 and multicomponent glasses. Why? This effect is explained (hypothetically) as a consequence of the thermal diffusion of sodium cations at high repetition rate of the laser pulses, but not demonstrated. A quantitative analysis of these regions by XPS and/or EDXS mapping is needed to clarify this supposition.

We thank the reviewer for taking the time to read and consider the manuscript and for appreciation of our study. Revised and newly added fragments of the text are tracked by the blue font.

Indeed, quantitative mapping of elemental composition in the modified glass area by XPS and/or EDXS is a quite reasonable way to check the suggested hypothetical mechanism. However, this sodium germanate glass is a rather difficult object for electron microscopy methods. Due to its high crystallization ability and rather low softening temperature, it easily degrades, changes its structure or crystallizes under the probe electron beam so it’s not easy to get a relevant picture of the laser-written structure. In particular, this is why we used atomic force microscopy rather than electron microscopy to visualize the submicron structure of the laser-written domains in this and the previous studies. Electron microscopy analysis is planned and now in progress but it will take some time and efforts to find the proper conditions or, perhaps, we’ll even have to change glass composition under study towards the lower content of sodium, which would increase the softening temperature and decrease crystallization ability. So we consider it to be a study worth of a separate publication.

Reviewer #2: The paper is clear and well-written, however, I have some minor observations to be addressed:

- Most of the experiments (except those devoted to the influence of the pulse duration), were performed with laser pulses of 1.2 ps, but described as femtosecond laser pulses. Please clarify this point.

This is a reasonable note because 1.2 ps value formally belongs to a picosecond range in the time scale. However, femtosecond laser pulses are usually regarded as having a duration in the range from a few femtoseconds to hundreds of femtoseconds (not a few thousands), while picosecond laser pulses - from a few picoseconds to hundreds of picoseconds. In terms of the interaction with the matter and of the laser-induced effects, 1.2 ps pulse has much more in common with sub-ps pulses of hundreds fs duration than with pulses having a duration of tens or hundreds ps.

That's why we suggest that addressing these pulses as femtosecond should be kept.

Reviewer #2: -  In the caption of Figure 4, „and 210 nJ” should be erased.

Thank you for catching this error. It is corrected now.

Reviewer #2: - There are some small English mistakes in the manuscript. Please check and correct them.

We checked English throughout the manuscript and corrected some mistakes and typos.

Round 2

Reviewer 2 Report

The authors have addressed all the suggestions I have made to improve the quality of the paper, so I recommend the publication.